# A Novel Environment Estimation Method of Whole Sample Traffic Flows and Emissions Based on Multifactor MFD

**DOI:** 10.3390/ijerph192416524

**Published:** 2022-12-09

**Authors:** Jinrui Zang, Pengpeng Jiao, Guohua Song, Zhihong Li, Tingyi Peng

**Affiliations:** 1Beijing Advanced Innovation Center for Future Urban Design, Beijing University of Civil Engineering and Architecture, Beijing 100044, China; 2Key Laboratory of Transport Industry of Big Data Application Technologies for Comprehensive Transport, Beijing Jiaotong University, Beijing 100044, China

**Keywords:** emission estimation, pattern clustering, pattern recognition, fundamental diagram, traffic volume estimation

## Abstract

Vehicle emissions seriously affect the air environment and public health. The dynamic estimation method of vehicle emissions changing over time on the road network has always been the bottleneck of air quality simulation. The dynamic traffic volume is one of the important parameters to estimate vehicle emission, which is difficult to obtain effectively. A novel estimation method of whole sample traffic volumes and emissions on the entire road network based on multifactor Macroscopic Fundamental Diagram (MFD) is proposed in this paper. First, the intelligent clustering and recognition methods of traffic flow patterns are constructed based on neural network and deep-learning algorithms. Then, multifactor MFD models are developed considering different road types, traffic flow patterns and weekday peak hours. Finally, the high spatiotemporal resolution estimation method of whole sample traffic volumes and emissions are constructed based on MFD models. The results show that traffic flow patterns are clustered efficiently by the Self-Organizing Maps (SOM) algorithm combined with the direct time-varying speed index, which describe 91.7% traffic flow states of urban roads. The Deep Belief Network (DBN) algorithm precisely recognizes 92.1% of the traffic patterns based on the speeds of peak hours. Multifactor MFD models estimate the whole sample traffic volumes with a high accuracy of 91.6%. The case study shows that the vehicle emissions are evaluated dynamically based on the novel estimation method proposed in this paper, which is conducive to the coordinated treatment of air pollution.

## 1. Introduction

Vehicle exhaust pollution is an important factor threatening public health [1]. Dynamic estimation of mobile pollution sources changing over time on the road network is the fundamental data for air pollution control and public health promotion [2]. Pollution sources are required to be controlled in coordination in air quality management [3]. Fixed pollution sources such as steel plants [4], power plants [5] and meteorological parameters [6] have been detected with high space-time resolution. However, the emissions of mobile vehicle sources still remain at the level of annual total amount measurement due to the limitation of emission detection technology [7]. The time and space granularity of vehicle emissions evaluation makes it difficult to measure it consistently in comparison with other pollution sources.

Traffic volume covering every link of the entire road network is one of the important parameters for dynamic emission estimation, which is difficult to collect based on the existing traffic detection with low coverage and is not estimated effectively [8]. The short-term traffic flow prediction method based on machine learning, which depends on huge historical traffic flow data, has been well studied [9]. Historical flow data of high-grade roads are sufficient, but the flow data of low-grade roads are extremely lacking [10]. Traffic flow characteristics extracted from historical laws based on machine learning reflect with difficulty the real-time traffic state on the road network, especially when accidental traffic events occur.

Although the whole sample traffic volumes of the entire road network are difficult to collect, the detection technology of speed data from the large-scale road network is well developed [11,12]. The traffic fundamental diagram model describes the relationship between speed and volume [13]. It is an effective method to estimate volume by the traffic fundamental diagram model based on speed data. The prediction accuracy for traffic volume based on the traffic fundamental diagram is improved with the increase of time granularity, which is relatively low when the time granularity is at the minute level and is relatively high when the time granularity is at the hour level [14]. This method is not widely used for short-term traffic flow prediction. The hourly granularity volume has met the demand for emission data. The traffic fundamental diagram is used to estimate the dynamic real-time flow based on the timely speed data in this paper.

The influencing factors of MFD are analyzed quantitatively, including road types, weather conditions, traffic patterns and travel periods. A novel estimation method of whole sample traffic flows and emissions based on multifactor MFD is proposed. The research results of this paper can be implemented for the preparation of a dynamic vehicle emission inventory on the road network, which is helpful for transportation and environmental departments to formulate policies of energy conservation and emission reduction. It plays a major role in promoting public health.

## 2. Literature Review

The quantitative estimation of dynamic emissions is fundamental for pollution control. Mobile sources have become an important part of air pollution, accounting for 55.6% of the total nitrogen oxide emissions in China in 2020 [15]. The Ministry of Ecology and Environment of China plans to evaluate emission sources with a uniform time granularity. Industrial, agricultural and living sources are fixed sources, which make it easy to obtain continuous dynamic data due to fewer pollution sources and simple monitoring technology [4,5,6]. Mobile sources have complex detection technologies and high testing costs, which remain at the level of static annual total quantity evaluation [16]. The evaluation granularity of vehicle emissions cannot be consistent with other pollution sources in terms of time and space, which limits the real-time detection and overall evaluation of air pollution. Therefore, dynamic estimation of vehicle emissions has always been a bottleneck problem in dynamic air quality simulation.

One of the important reasons why it is difficult to estimate dynamic vehicle emissions is the lack of an effective method to estimate whole sample dynamic traffic volumes of urban road networks. Vehicle emissions can be calculated based on single vehicle emission factors and vehicle kilometers traveled (VKT) [14]. The single vehicle emission factor is the emissions per kilometer of a single vehicle, which has a good research foundation and can be effectively obtained [17]. The whole sample dynamic traffic volume is the traffic volume that dynamically changes over time on every road link of a road network, which is an important parameter to calculate the dynamic VKT [16]. The main sources of traffic volume in existing studies include field traffic surveys, traffic simulations and traffic flow models. Field traffic surveys include fixed detector detection and manual surveys, which are costly and difficult. The road network coverage of field traffic survey data is limited, and it is difficult to obtain large-scale real-time dynamic data [18]. The traffic simulation method [19] has difficulty ensuring the consistency of the simulation environment and the actual traffic conditions. The traffic flow model method uses speed data to estimate volume based on traffic fundamental diagrams [20]. Dynamic speed data of large-scale road networks can be easily obtained based on massive data, such as floating car data (FCD) [11] and travel trajectories collected from mobile applications [21], which have high temporal and spatial continuity. Dynamic traffic control, such as signal timing optimization, requires small time granularity (five minutes) and high estimation accuracy of traffic volume [22]. The estimation accuracy of small time granularity traffic volume calculated using a fundamental diagram model cannot meet the requirement of the traffic control field; therefore, a fundamental diagram is not widely used in the traffic control field [16]. The hourly time granularity volume meets the emission estimation requirement and can be estimated with high accuracy based on a traffic flow fundamental diagram [14]. This paper proposes a method to calculate whole sample dynamic volumes of urban road networks based on traffic flow fundamental diagrams.

The traffic fundamental diagram was originally proposed by Greenshields in the form of a linear model in the 1930s [23]. Then, the logarithmic, exponential and vehicle-following models were separately introduced by Greenberg [24], Underwood [25] and Pipes [26]. In 1995, the Van Aerde model was proposed [27] with four parameters, a single structure, continuous feature and flexible calibration. It was found that the Van Aerde model fits the traffic flow of highways and expressways well [28]. The Underwood model has good performance in describing major arterial road and minor arterial road traffic flows [20]. However, the data used to develop these models are mostly collected from a single observation point. With the development of urban road networks and increasing traffic flow complexity, traffic fundamental diagrams have gradually expanded from single-section to multi-section networks. The concept of a macroscopic fundamental diagram (MFD) was proposed [29], and the existence of MFDs was verified based on field survey data and simulation data [30,31]. Some scholars [32,33] proposed new fundamental traffic diagrams based on a MFD. Existing research on MFDs is mainly based on simple structure road networks, excellent traffic conditions or simulation methods. MFD models based on real traffic conditions and complex road networks need to be further studied.

Traffic flow change patterns are divided to improve the effectiveness and accuracy of traffic volume prediction. It was pointed out that the travel behaviors of residents are cyclical and repetitive [18], which led to multiple but limited patterns of traffic flow patterns. Representative traffic flow patterns were constructed based on traffic volume and speed data in existing studies as follows. Jia et al. [34] proposed a spatiotemporal neural network model to predict traffic flow for each road segment. The traffic flow was divided into recent, daily and weekly parts. A multimode traffic flow prediction method with clustering-based attention convolution LSTM was proposed by Huang et al. [35] to model spatial-temporal data of traffic flow by integrating weather, wind speed, holidays and other factors to improve the prediction accuracy. Ma et al. [36] proposed a dynamic time warping method to select the appropriate historical data for daily traffic flow forecasting based on daily traffic flow pattern influence factors, such as season, day of the week, weather and holiday. The prediction effectiveness and accuracy of traffic volume are improved by dividing traffic flow patterns. Field survey traffic flows on highways or expressways are used as clustering indexes in most studies, which do not take into account the influence of the geographic location, road type and land use function on the division of traffic flow patterns. It is difficult to discover the differences among various traffic flow change patterns. A traffic flow clustering method considering various factors is proposed based on various road types of traffic flow data in this paper.

Above all, traffic flows can be clustered into limited typical patterns due to the regularity of travel and the grading characteristics of roads. Predicting flows for different patterns separately is an effective method to improve the prediction accuracy. Although whole sample dynamic traffic volumes of urban road networks are difficult to obtain, the large-scale speed data of a road network can be obtained based on a floating car system. The traffic flow model is used in this paper to predict traffic volume based on speed data for different patterns.

## 3. Methodology

### 3.1. Research Steps

The general methodology of this study is divided into the following steps, as shown in Figure 1.

First, the data preparation of traffic speed and volume for developing traffic flow patterns and traffic fundamental diagram models is introduced. The speed and volume data are aggregated into a 1-h granularity. Then, the time and location information of different data sources are matched.

Second, a clustering method of traffic flow patterns based on multidimensional features is constructed. The optimal traffic flow pattern clustering method is designed using different clustering methods. A library of typical traffic flow patterns under various traffic conditions is constructed.

Third, a traffic flow pattern recognition method based on speed indexes in different periods is constructed. Indexes suitable for different traffic application scenarios are designed to recognize traffic flow patterns rapidly.

Finally, a method is proposed to estimate whole sample dynamic traffic volumes of urban road networks based on clustering and recognition of traffic flow patterns. Multifactor MFD models are constructed based on road types, traffic flow patterns and morning and evening peak hours on weekdays. The dynamic traffic volume is estimated based on real-time speed data. The dynamic emissions of a road network before and after the traffic restriction policy are calculated as a case study.

### 3.2. Data Source and Preparation

The data used in this study include three parts.

(1) The average spatial speed data are from FCD in 5-min intervals of various road types in Beijing. The number of days for highway, expressway, major arterial road and minor arterial road data were 164, 175, 265 and 213, respectively, in 2018. Data on weekdays, weekends and holidays under different weather conditions are included. The attributes of average spatial speed data include link ID, road name, date, time, road type and average spatial speed.

(2) The traffic volume data from remote traffic microwave sensors (RTMSs) in 2-min intervals of various road types in Beijing are used. The number of days for highway, expressway, major arterial road and minor arterial road data were 132, 144, 205 and 153, respectively, in 2018. The attributes of average spatial speed data include detector number, road name, date, time and traffic volume.

(3) The traffic volume from traffic survey data at 5-min intervals under various road types in Beijing is used. The RTMS data on highways and expressways are relatively abundant, whereas the data on major arterial roads and minor arterial roads are relatively lacking. The traffic volume survey data are used to supplement the RTMS volume data. The attributes of traffic survey data include survey date, road name, vehicle type and traffic volume.

After data quality control, the speed data are integrated according to Equation (1).
(1)$$V_{j}=\frac{\sum_{i}^{n}v_{i}}{n}$$
where $V_{j}$ is the speed of hour *j* in 1-h intervals, *v*_*i*_ is the original (60/n) min speed in each 1-h interval and *n* is the number of the original (60/n) min data in each 1-h interval. The traffic volume data are integrated according to Equation (2).
(2)$$Q_{j}=\sum_{i}^{n}q_{i}$$
where is the traffic volume of hour *j* in 1-h intervals and $q_{i}$ is the original (60/n) min traffic volume in each 1-h interval.

Then, the speed and traffic data are normalized using Equation (3) to avoid the effect of the quantity of the data.
(3)$$x_{i}^{*}=\frac{x_{i}-x_{min}}{x_{max}-x_{min}}$$
where $x_{i}$ is the speed (km/h) or traffic volume (pcu/h/lane), $x_{i}^{*}$ is the normalized speed or traffic volume and $x_{min}$ and $x_{max}$ are the minimum and maximum values of all records in a day, respectively.

## 4. Speed Clustering Method

The pattern clustering method of speeds includes the following processes: (1) clustering algorithm design; (2) clustering index selection; and (3) clustering effect evaluation.

### 4.1. Traffic Speed Pattern Definition

Affected by differences in the road function and travel demand, speeds on different roads and dates show different change patterns. The travel purpose on weekdays is mostly commuting, and the travel time is relatively fixed. The speed presents a bimodal distribution of morning and evening peaks with fixed time. The commuting demand decreases, whereas the entertainment demand increases on weekends and holidays. The traffic speed changing patterns of weekends and holidays differ from those of weekdays. The representative form of the traffic flow change curves over time is defined as the traffic flow pattern in this paper.

### 4.2. Determination of the Optimal Number of Clusters

A method to automatically determine the optimal number of clusters is proposed. The silhouette coefficient (*SC*) is adopted to determine the number of categories. The silhouette coefficients of different clustering number of speed curves are automatically calculated based on the K-means clustering algorithm. The optimal clustering number is determined when the profile coefficient reaches the maximum.

The *SC* is used to evaluate the tightness of samples within the categories and the separation of samples between categories, as shown in Equation (4).
(4)$$Sil(a)=\frac{a_{a}-b_{a}}{\max\left\{a_{a},b_{a}\right\}}=\frac{\underset{K^{\prime }\neq K(a)}{\min}\left\{{\mathrm{mean}}_{A_{c}\in K^{\prime }}\left|A_{a}A_{c}\right|\right\}-{\mathrm{mean}}_{A_{b}\in K(a)}\left|A_{a}A_{b}\right|}{\max\left\{\underset{K^{\prime }\neq K(a)}{\min}\left\{{\mathrm{mean}}_{A_{c}\in K^{\prime }}\left|A_{a}A_{c}\right|\right\},{\mathrm{mean}}_{A_{b}\in K(a)}\left|A_{a}A_{b}\right|\right\}}$$
where *Sil*(*a*) is the silhouette of *A_a_*, *a_a_* is the minimum value of the average distance between sample *A_a_* and the samples in other categories, *b_a_* is the average distance between sample *A_a_* and other samples in the same category, *A_b_* is a sample in the same category of *A_a_*, *A_c_* is a sample of different categories from *A_a_*, *K*(*a*) is the category of *A_a_* and *K’* is a different category from *K*(*a*).

### 4.3. Clustering Algorithms

The hierarchical clustering method, K-means clustering method and SOM neural network clustering method are used to classify the speed time-varying pattern, which proves to be effective in clustering traffic flows.

#### 4.3.1. Hierarchical Clustering Method

The solution of hierarchical clustering method is carried out hierarchically, which is simple and intuitive. The pseudo code of hierarchical clustering algorithm is described as follows [37].

Step 1: Enter *N* cluster samples and the number of expected clusters *K*. Each sample is regarded as a cluster, and the distance *d_i j_* between every two clusters is calculated to obtain the initial clustering matrix, as shown in Equation (5).
(5)$$d_{ij}=\sqrt{\overset{p}{\underset{k=1}{\sum }}{\left(x_{ik}-x_{jk}\right)}^{2}}$$
where, *d_i j_* is the Euclidean distance between sample point *i* and sample point *j*; *p* is the dimensions of clustering index; $x_{ik}$ and $x_{jk}$ are the coordinates of point *i* and point *j* respectively.

Step 2: Merge the two clusters with the smallest *d_i j_* into a new cluster.

Step 3: Recalculate the distance *d_ij_* between the new cluster and other clusters;

Step 4: Repeat Step 2 and Step 3 until the number of clusters becomes *K*.

#### 4.3.2. K-Means Clustering Method

K-means clustering method converges fast and has a better clustering effect when the sample gap between classes is obvious. The pseudo code of K-means clustering algorithm is described as follows [38].

Step 1: Enter *N* cluster samples and the number of expected clusters *K*. *K* samples are randomly selected from all samples as the initial cluster center.

Step 2: Calculate the Euclidean distance from each sample to every cluster center and classify it with the nearest center.

Step 3: Recalculate the average coordinate of each cluster as the new cluster center.

Step 4: Calculate the square error *E*(*t*) of all sample points and compare it with the previous error *E*(*t* − 1).

Step 5: If *E*(*t*) − *E*(*t* − 1) < 0, turn to step 2, otherwise the algorithm ends.

#### 4.3.3. SOM Clustering Method

The SOM neural network was proposed by Professor Kohonen [39]. The SOM network includes an input layer and output layer, which are connected by weight vectors. The input layer is a high-dimensional vector, and the output layer is a series of ordered nodes organized on a two-dimensional grid. The clustering results of SOM are highly visual and interpretable. SOM clustering requires the following steps [40]:

Step 1: Initialize the network and assign the initial value to each node weight of the output layer.

Step 2: Randomly select the input vector from the input samples to find the weight vector with the minimum distance from the input vector.

Step 3: Define the winning unit and adjust the weight in the adjacent area of the winning unit to make it close to the input vector.

Step 4: Provide new samples and conduct training.

Step 5: Shrink the neighborhood radius, reduce the learning rate, repeat until it is zero and output the clustering results.

### 4.4. Clustering Index Design

Clustering index refers to the feature vector that is related to the clustering analysis target, reflects the characteristics of the classification object and has obvious differences in different objects. Three indexes are designed in this study: (1) direct index (*DI*) based on time-varying speed differences, (2) comprehensive index (*CI*) based on speed statistical characteristics and curve geometric characteristics, and (3) simplified index (*SI*) based on speed statistical characteristics and curve geometric characteristics.

#### 4.4.1. Direct Index Based on Time-Varying Speed Differences

The speeds of 0:00–24:00 at 1-h granularity are used as a clustering index to describe the difference in the traffic flow at different times directly. The speed of different time periods has different abilities to describe the characteristics of traffic flow. The speeds of peak hours can reflect the difference in traffic flow better than speeds of early morning. The coefficient of variation of each hour speed is used as the weight in this study. A larger weight is given to speeds with more different characteristics to improve the discrimination of traffic flows on different days.

The coefficient of variation calculation method is shown in Equation (6):(6)$$CV_{j}=\frac{\sigma_{j}(x)}{E_{j}(x)}=\frac{\sum x_{ij}/m}{\sqrt{\sum {\left[x_{ij}-E_{j}(x)\right]}^{2}/m}}$$where $CV_{j}$ is the coefficient of variation of the initial index of *j*,
$\sigma_{j}(x)$ is the population standard deviation of the initial index of all samples *j* to be clustered, *E_j_*(*x*) is the mean value of *j* initial index values of all samples to be clustered, *x_ij_* is the initial index value of *j* in sample *i* and *m_j_
*is the number of samples to be clustered.

The calculation method of the clustering index is shown in Equation (7):(7)$$a_{ij}=x_{ij}\times CV_{j}$$where $a_{ij}$ is the final clustering index value of *j* in sample *i*. The final input cluster sample index matrix is shown in Equation (8):(8)$$\left[\begin{array}{cccc} a_{11} & a_{12} & \cdots & a_{1n}\\ a_{21} & a_{22} & \cdots & a_{2n}\\ \vdots & \vdots & \ddots & \vdots \\ a_{m1} & a_{m2} & \cdots & a_{mn} \end{array}\right]$$
where *n* is the number of indexes. In the DI, *n* is 24.

#### 4.4.2. Comprehensive Index Based on Speed Statistical Characteristics and Curve Geometric Characteristics

The *CI* includes statistical characteristics of speeds and geometric characteristics of the speed time-varying curve. Statistical characteristics of speeds include the maximum speed, the minimum speed, the ratio of the maximum speed and the minimum speed, the average speed of different time periods, such as 0:00 to 24:00, 0:00 to 6:00, 20:00 to 23:00, 10:00 to 12:00, 2:00 to 4:00, morning peak hours, evening peak hours, the ratio of the morning peak hours speeds and the evening peak hours speeds and the variance of speeds of 0:00–24:00. The time-varying speed curve is divided into concave and convex shapes: (1) concave shape from 6:00 to 12:00; (2) concave shape from 12:00 to 22:00; and (3) convex shape from 10:00 to 14:00, as shown in Figure 2. The geometric characteristics of the speed time-varying curve include the kurtosis and skewness of each concave and convex shape. The comprehensive indexes are shown in Table 1.

#### 4.4.3. Simplified Index Based on Speed Statistical Characteristics and Curve Geometric Characteristics

There are problems in the *CI*, such as the excessive number of indexes, correlation between indexes and different capabilities to describe traffic flow characteristics. The *SI* is constructed using the principal component analysis method to extract the most representative key indexes from the *CI*, which can effectively classify traffic flow, simplify the feature matrix and improve the calculation efficiency. The *SI* are shown in Table 2.

### 4.5. Clustering Effect Evaluation Indexes

The clustering effect evaluation indexes used in this paper are the silhouette coefficient (*SC*), consistency within a category, differences between categories and the coverage ratio (*CR*).

#### 4.5.1. Consistency within a Category

The consistency within a category is the coefficient of variation (*CV*) of samples within a category, which is used to measure the consistency of speeds within a category, as shown in Equation (9).
(9)$$CV_{\mathrm{in}}=\frac{1}{n}\sum_{k=1}^{n}CV_{\mathrm{in}}^{k}=\frac{1}{n}\sum_{k=1}^{n}(\frac{1}{l}\sum_{i=1}^{l}CV_{i,\mathrm{in}}^{k})$$
where $CV_{\mathrm{in}}$ is the average *CV* within a category, $CV_{\mathrm{in}}^{k}$ is the *CV* within category *k*, *n* is the number of clusters, $CV_{i,\mathrm{in}}^{k}$ is the *CV* within a category at time *i* in category *k* and *l* is the number of time intervals in a day.

#### 4.5.2. Difference between Categories

The difference between categories is the coefficient of variation of samples between categories, which is used to measure the differences between categories, as shown in Equation (10).
(10)$$CV_{\mathrm{out}}=\frac{1}{l}\sum_{i=1}^{l}CV_{i,\mathrm{out}}$$
where $CV_{\mathrm{out}}$ is the *CV* of the speed of all categories and $CV_{i,\mathrm{out}}$ iis the *CV* s between categories at time *i*.

#### 4.5.3. Coverage Ratio

The coverage ratio is the proportion of reasonably classified samples to the total samples in a category, which reflects the rationality of the clustering results and the representativeness of the traffic flow pattern. A reasonably categorized sample is defined as the average Pearson correlation coefficient between the sample and other samples within a category greater than 0.95. The coverage ratio is calculated by Equation (11).
(11)$$CR=\frac{D_{1}}{D}\times 100\%$$
where *D*_1_ is the number of correctly classified days and *D* is the total number of days.

#### 4.5.4. Clustering Results Evaluation

The silhouette coefficients, consistencies within a category, differences between categories and coverage ratios of traffic flow patterns under different indexes are shown in Figure 3.

As shown in Figure 3, the categories of the *DI* have the highest $CV_{\mathrm{out}}$ and CR and have higher *SC*. The differences between samples of different categories are high, and the proportion of samples reasonably classified reaches 91.7%. The categories of the *CI* have higher *CR*, but lower *SC* and $CV_{\mathrm{out}}$. The difference between samples of different categories is lower than *DI* and *SI*. The categories of the *SI* have higher *SC* and $CV_{\mathrm{out}}$. The difference between samples of different categories is great. However, the categories of the *SI* have the lowest *CR*. The proportion of samples reasonably classified is lower than *DI* and *CI*. Above all, *DI* has the best clustering effect, which has the strongest ability to classify the samples reasonably and distinguish the differences between different categories.

The categories of *DI* based on different algorithms are shown in Figure 4.

The hierarchical clustering algorithm based on *DI* clusters speed curves into ordinary weekdays and weekends, rainy days, non-congestion weekdays, days in lieu, The Dragon Boat Festival, festivals, prominent evening peak and prominent evening peak patterns. There is no distinction between weekdays and weekends, and the samples of some patterns are too few to be practical based on hierarchical clustering algorithm and the *DI* method.

The K-means clustering algorithm based on *DI* clusters speed curves into Mondays to Thursdays, Fridays, weekends, rainy days, festivals, weekends and festivals, prominent evening peak and prominent evening peak patterns. The categories of weekends and festivals are not clear. The common features of the samples in some categories are not obvious.

The SOM clustering algorithm based on *DI* clusters speed curves into Mondays to Thursdays, Fridays, Saturdays, Sundays, rainy days, festivals, prominent evening peak and prominent morning peak patterns. The traffic flow patterns are reasonable and practical, which have the following characteristics: (1) the differences between categories are the highest; (2) the traffic patterns are more detailed, and factors such as weather conditions are taken into account to better reflect the characteristics of different traffic flow patterns.

Above all, the SOM neural network clustering algorithm based on the *DI* is the most effective clustering method, which is used to cluster the traffic flow. The speed patterns and volume patterns on an expressway are shown in Figure 5 and Figure 6.

## 5. Traffic Flow Pattern Recognition Method

### 5.1. Traffic Flow Pattern Recognition Algorithms

Pattern recognition algorithms based on intelligent algorithms have good performance in the traffic flow field. A back propagation (BP) neural network is widely used because of the simplicity and flexibility of the structure. An optimization algorithm is used to improve the slow convergence and easily fall into the extreme value of a BP neural network, such as a genetic algorithm (GA) and a simulated annealing genetic algorithm (SAGA). A DBN has the ability to extract high-dimensional abstract features of traffic flows through a hidden multilayer structure, which recognizes the pattern of large-scale data efficiently. Therefore, the BP, GA-BP, SAGA-BP and DBN algorithms are used to recognize the traffic flow patterns [41,42,43,44]. The flow chart of each algorithm is shown in Figure 7. The parameters of the algorithms are shown in Table 3.

### 5.2. Traffic Flow Pattern Recognition Indexes

A variety of traffic flow pattern recognition indexes are designed according to traffic flow characteristics to adapt to different collection conditions of traffic data and traffic application scenarios.

(1)Index I: Speeds from 0:00 to 24:00

Speeds from 0:00 to 24:00 contain the complete traffic flow characteristics over time, which are suitable for pattern recognition of historical traffic flow data, such as the evaluation of effect on traffic policy.

(2)Index II: Speeds from 0:00 to 6:00

Speeds from 0:00 to 6:00 describe the characteristics of the early morning traffic flow, which are suitable for real-time traffic scenarios.

(3)Index III: Speeds from 0:00 to 12:00

Speeds from 0:00 to 12:00 describe the traffic flow characteristics in the early morning and morning peak hours, which have a degree of real-time performance.

(4)Index IV: Speeds from 6:00 to 10:00

Speeds from 6:00 to 10:00 describe the traffic flow characteristics of morning peak hours, which have a degree of real-time performance. Index IV has fewer dimensions, which is useful to improve the efficiency of pattern recognition.

(5)Index V: Speeds from 6:00 to 10:00 and 17:00 to 21:00

Speeds from 6:00 to 10:00 and 17:00 to 21:00 describe the traffic flow characteristics of morning and evening peak hours, which have a better ability to reflect the difference in traffic flow characteristics of different categories.

The pattern recognition results of different algorithms based on each index are analyzed on different road types, as shown by taking an expressway as an example in Figure 8.

### 5.3. Pattern Recognition Evaluation Indexes

The evaluation indexes of the pattern recognition effect include the recognition rate based on a confusion matrix (CM) and the pattern recognition simulation time. The recognition rates and simulation times of different algorithms based on each index are shown in Table 4.

The results of pattern recognition of the different algorithms based on each index are analyzed on different road types, and the conclusions are as follows.

(1) The accuracy and efficiency of pattern recognition of the DBN are the highest under various conditions. The GA-BP and SAGA-BP have better performance than the BP algorithm.

(2) The index of speeds from 0:00 to 24:00 has the best recognition effect, and the average correct recognition rate of all road types is 94.87%. The index of speeds from 6:00 to 10:00 and 17:00 to 20:00 has a good recognition effect, and the average recognition of all road types is 93.26%.

(3) The average recognition rates of the index of speeds from 0:00–12:00 and the index of speeds from 6:00–10:00 exceed 80%, and the highest average recognition rate is 89.39%.

(4) The index of speeds from 0:00 to 6:00 has a low recognition rate, and the correct recognition rate of all road types is 59.36%. The early-morning data in each category have relatively small differences, which are not suitable as indexes of traffic flow patterns.

## 6. Traffic Flow Estimation Method

The traffic flow estimation method is proposed based on the traffic flow fundamental diagram. The influencing factors on the fundamental diagram are analyzed, and then the fundamental diagrams are constructed for different conditions.

### 6.1. Analysis of the Factors Influencing the Fundamental Diagram

The traffic flow characteristics are used to develop the traffic flow fundamental diagram based on traffic flow theory, such as the traffic composition, the morning and evening peak and the weather conditions.

#### 6.1.1. Traffic Composition

Traffic composition is the proportion of different vehicle types in the traffic flow. Large vehicles occupy more road space than small passenger vehicles. The ability of large vehicles to accelerate and decelerate is lower than that of small passenger cars. Large vehicles have a larger headway [45]. The proportion of large vehicles is an important factor affecting traffic capacity.

The large-vehicle proportions in different traffic flow patterns are calculated based on the traffic flow data of the North Second Ring Road, North Third Ring Road and North Fourth Ring Road, as shown in Figure 9. The large-vehicle proportion from 0:00 to 6:00 in each pattern is higher, and it is above 20% in each hour except on Sundays. The large-vehicle proportion is below 15% each hour after 7:00 with low volatility. There are differences in the large-vehicle proportion among different patterns, and the large-vehicle proportion on weekends is lower than that on weekdays. Due to the difference in the large-vehicle proportion, there are differences among the parameters of the traffic pattern.

To analyze the influence of the large-vehicle proportion on traffic flow parameters, fundamental diagrams of different large-vehicle proportions at intervals of 10% are developed in this study, as shown in Figure 9. It was found that the traffic capacity decreases as the large-vehicle proportion increases. It is necessary to establish the fundamental diagrams for different patterns and different time periods.

#### 6.1.2. Morning and Evening Peak Hours

The fundamental diagrams in the morning and evening peak hours are developed, taking expressways as an example. The RTMS data and FCD data of 15 sections of the Beijing Expressway from February to October 2018 are used to develop fundamental diagrams. To ensure that there are enough data, the data from 6:00 to 9:00 are selected as the morning peak hours, and the data from 17:00 to 20:00 are selected as the evening peak hours to build fundamental diagrams, as shown in Figure 9. The traffic capacity of morning peak hours is 1600 pcu/h, which is 3.75% larger than that in the evening peak hours. The travel purpose of travelers in the morning peak hours is commuting, which has high driving behavior consistency and traffic flow stability.

To analyze whether the differences in traffic capacity during morning and evening peak hours are consistent on different weekdays, a week of fundamental diagrams of morning and evening peak hours are developed on expressways, as shown in Figure 10. The traffic capacity in the morning peak hours is on average 3.24% higher than that in the evening peak hours, which is consistent and repetitive on different weekdays.

#### 6.1.3. Weather Conditions

It is pointed out in the Highway Capacity Manual [41] that bad weather conditions reduce road speed and capacity compared with normal weather conditions. The data of rainy days on 17 July 2018, and normal days of the same month are used to analyze the influence of weather conditions on the speed and fundamental diagrams, as shown in Figure 11.

In summary, the road type, traffic composition, travel purpose and weather conditions are all factors influencing traffic flow fundamental diagrams, which are different under different traffic flow patterns. Therefore, it is necessary to construct fundamental diagrams for different traffic flow patterns and weekday peak hours.

### 6.2. Fundamental Diagram Models

The traffic flow fundamental diagrams of highways and expressways are constructed based on the Van Aerde model. The traffic flow fundamental diagrams of major arterial roads and minor arterial roads are constructed based on the Underwood model. The expressway is taken as an example to introduce the MFD model construction method. There are 142 days of expressway flow and speed data selected in this study, among which 85 days of data are used to construct MFD models, and 57 days of data are used to analyze the model uncertainty. The fundamental diagrams for different road types, traffic flow patterns and peak hours of workdays are developed for three proposals, as shown in Figure 12. Traffic flow characteristic parameters of different proposals on expressways are in Table 5.

#### 6.2.1. Proposal I: Fundamental Diagrams for Different Road Types

Proposal I is to develop fundamental diagrams for different road types. The fundamental diagrams on expressways are shown in Figure 12a. The characteristic parameters on expressways for Protocol I are shown in Table 5.

#### 6.2.2. Proposal II: Fundamental Diagrams for Different Traffic Flow Patterns

Proposal II is to develop fundamental diagrams for different traffic flow patterns and road types based on the traffic flow patterns constructed in this study, as shown in Figure 12b. The characteristic parameters for Protocol II are shown in Table 5.

The fundamental diagram of each pattern is different. There are lower large-vehicle proportions and higher traffic capacities on weekends and holidays than on weekdays. The traffic capacity on rainy days is lower than that on days of normal weather. The traffic capacity on Saturday, Sunday and holidays patterns is 2.61%, 3.58%, 5.21% higher than that in the Monday to Thursday weekday pattern, respectively, and 3.28%, 4.26%, 5.90% higher than that in Friday weekday pattern. The capacity of the rainy-day pattern is reduced by 6.84% and 6.23% compared with the Monday to Thursday weekday pattern and the Friday weekday pattern, respectively.

#### 6.2.3. Proposal III: Fundamental Diagrams for Morning/Evening Peak Hours of Weekday Patterns

Proposal III is to develop the fundamental diagrams for peak hours of weekday patterns and road types, as shown in Figure 12c. The characteristic parameters for Protocol III are shown in Table 5.

The capacity of morning peak hours is higher than that of the evening peak hours in each pattern. The capacity of morning peak hours of the Monday to Thursday weekday pattern and Friday weekday pattern is 3.66% and 4.11% higher than that of evening peak hours, respectively.

The fundamental diagrams of Proposal I, Proposal II and Proposal III on the Beijing highways, expressways, major arterial roads and minor arterial roads are developed in this study.

#### 6.2.4. Error Analysis

The RTMS and FCD data that have not been used for the model construction of 52, 57, 93 and 74 days on Beijing highways, expressways, major arterial roads and minor arterial roads are selected to estimate the accuracy of the MFD models under the three proposals. The FCD speed data are used to estimate the volume data based on the fundamental diagrams. The errors from 6:00 to 24:00 between the fundamental diagrams estimated volumes and the field measured volumes are calculated, as shown in Table 6.

The prediction accuracy of the fundamental diagrams for different weekday peak hours and road types from 6:00 to 24:00 are improved. The average relative errors of Proposal I, Proposal II and Proposal III on the highway are 15.76%, 11.82% and 9.76%, respectively, and the average relative errors of Proposal III and Proposal II are 6.06% and 3.94% lower than that of Proposal I, respectively. The average relative errors of Proposal I, Proposal II and Proposal III on the expressway are 12.65%, 8.88% and 7.64%, respectively, and the average relative errors of Proposal III and Proposal II are 5.01% and 3.77% lower than that of Proposal I, respectively. The average relative errors of Proposal I, Proposal II and Proposal III on the major arterial road are 14.45%, 11.09% and 8.58%, respectively, and the average relative errors of Proposal III and Proposal II are 5.87% and 3.36% lower than that of Proposal I, respectively. The average relative errors of Proposal I, Proposal II and Proposal III on the minor arterial road are 14.35%, 8.74% and 7.81%, respectively, and the average relative errors of Proposal III and Proposal II are 6.54% and 5.61% lower than that of Proposal I, respectively.

Above all, Proposal III has the highest accuracy. Representative roads of different road types on 3 April 2018 are selected to further evaluate the accuracy of the fundamental diagrams of Protocol III. The estimated volume of Protocol III is compared with the measured RTMS volume, as shown in Figure 13.

The average relative error of the Beijing-Kaisha highway is 7.19%, and the average absolute error of that road is 48 pcu/h. The average relative error of the North Fourth Ring Expressway is 7.47%, and the average absolute error of that road is 49 pcu/h. The average relative error of the Fangzhuang major arterial road is 6.83%, and the average absolute error of that road is 32 pcu/h. The average relative error of the Guang’an minor arterial road is 7.25%, and the average absolute error of that road is 16 pcu/h. The accuracy of volumes calculated by the fundamental diagrams of Protocol III constructed in this paper is relatively high.

### 6.3. Traffic Flow Estimation Method of the Whole Sample on the Urban Road Network

Fundamental diagrams for road types, traffic flow patterns, and weekday peak hours are developed in this study. Whole sample dynamic traffic volumes of the urban road network are calculated based on the real-time FCD data. The calculation steps for whole sample dynamic traffic volumes of urban road network are as follows:(1)The speed data input and quality control.(2)The map data input: including the road type, the number of lanes, the road length, etc.(3)The speed data and map data matching: the speed is matched with the map data.(4)Pattern recognition: the input speed is used for traffic flow pattern recognition based on the typical traffic flow pattern library and traffic flow pattern recognition method established in this study.(5)Fundamental diagram selection and flow calculation: the fundamental diagram selection based on traffic flow pattern and time information, flow data calculation and quality control.

The traffic flow of each road link can be calculated based on the traffic flow estimation method, which can be further used to calculate the dynamic vehicle emissions of the urban road network.

## 7. Case Study

The traffic restriction implementation effect of diesel trucks of the national stage III emission standard is evaluated as a case study. The changes in dynamic vehicle emissions before and after the implementation of the traffic restriction policy are calculated.

The traffic restriction policy of diesel trucks of the national stage III emission standard was published on 1 December 2018. A diesel truck of the national stage III emission standard is forbidden to drive within the Fifth Ring Road (excluding) from 6:00 to 23:00 every day. Trucks are forbidden to drive over 8 tons (including) on the Fifth Ring Road. Data from weekdays on 5 November, 6 November and 7 November 2018, before the implementation of the policy, and 3 December, 4 December and 5 December 2018, after the implementation of the policy, are used to calculate the emissions within the Fifth Ring Road.

The calculation method of the road link emissions is shown in Equation (12):(12)$$E_{i}=\sum EF_{j}^{R,v}\times Q_{i}\times L_{i}\times P_{j}^{R}$$where *E*_*i*_ is the emission of road link *i* (ton); $EF_{j}^{R,v}$ is the emission factor (g/km) of vehicle type *j* when its speed is *v* on road type *R*, $Q_{i}$ is the traffic volume of road link *i* (pcu/h); is the length of road link *i* (km); and $L_{i}$ is the proportion of vehicle type *j* on road type *R*. The vehicle emissions of the road network were the total emissions of each road link.

The emissions on the road network of each pollutant before and after the implementation of the traffic restriction policy in different periods is calculated, such as from 0:00 to 24:00, 6:00 to 23:00, 7:00 to 9:00, 17:00 to 19:00 and 20 to 22:00, as shown in Table 7.

The emissions of each pollutant on the whole road network within the Fifth Ring Road all decrease after the implementation of the traffic restriction policy from 6:00 to 23:00. The CO_2_ emissions decrease from 36,817.12 tons to 34,144.84 tons, with a decrease rate of 7.26%. The NO_x_ emission decreases from 135.96 tons to 118.05 tons, with a decreasing ratio of 13.18%. The CO emission decreases from 522.25 tons to 495.58 tons, with a decreasing ratio of 5.11%. The PM emissions decrease from 1.95 tons to 1.57 tons, with a decreasing ratio of 19.78%. The HC emission decreases from 96.56 tons to 91.56 tons, with a decreasing ratio of 5.18%.

The emission intensity on the road network of each pollutant before and after the implementation of the traffic restriction policy in different periods is calculated, such as from 0:00 to 24:00, 6:00 to 23:00, 7:00 to 9:00, 17:00 to 19:00 and 20 to 22:00, as shown in Table 8.

The emission intensity on the road network of each pollutant before and after the implementation of the traffic restriction policy in different periods is calculated, such as from 0:00 to 24:00, 7:00 to 9:00, 17:00 to 19:00 and 20 to 22:00 before and after the implementation of the traffic restriction policy is shown on the GIS map in Figure 14.

The emission intensities of CO_2_, NO_x_, CO, HC and PM in the road network all decrease after the traffic restriction. The emission intensities of CO_2_, NO_x_, CO, HC and PM decreased by 5.50%, 11.48%, 3.28%, 3.33% and 19.05%, respectively, from 0:00 to 24:00. The emission intensities of CO_2_, NO_x_, CO, HC and PM decreased by 6.03%, 13.08%, 3.48%, 3.55% and 21.44%, respectively, from 6:00 to 23:00. The emission intensities of CO_2_, NO_x_, CO, HC and PM decreased by 27.91%, 59.52%, 14.03%, 15.68% and 73.76%, respectively, from 2:00 to 4:00. The emission intensities of CO_2_, NO_x_, CO, HC and PM decreased by 3.12%, 7.84%, 1.20%, 1.43% and 14.89%, respectively, from 7:00 to 9:00. The emission intensities of CO_2_, NO_x_, CO, HC and PM decreased by 5.53%, 10.66%, 3.26%, 3.31% and 18.16%, respectively, from 10:00 to 12:00. The emission intensities of CO_2_, NO_x_, CO, HC and PM decreased by 6.52%, 11.37%, 4.57%, 4.43% and 18.27%, respectively, from 17:00 to 19:00. The emission intensities of CO_2_, NO_x_, CO, HC and PM decreased by 6.53%, 14.46%, 3.89%, 3.93% and 22.50%, respectively, from 20:00 to 22:00.

## 8. Conclusions

The main research results and conclusions of this paper are as follows:

(1) The SOM clustering method based on the direct indexes of time-varying speed differences effectively clusters urban traffic flow. Traffic flow patterns describe more than 90% of the traffic flows on urban road networks, such as Monday to Thursday, Friday, Saturday, Sunday, rainy days, holidays, prominent evening peak and prominent morning peak patterns.

(2) The recognition accuracy and efficiency of the DBN are the highest under various conditions. The GA-BP and SAGA-BP have better performance than the BP algorithm. The average correct recognition rates of different road types of speeds from 0:00 to 24:00, 0:00 to 12:00, 6:00 to 10:00, 6:00 to 10:00 and 17:00 to 21:00 are 94.87%, 82.45%, 80.96%, 93.26%, respectively. The early-morning data in each category are not suitable for traffic flow pattern recognition.

(3) In addition to the large-vehicle proportion and weather conditions identified by existing studies, it was found that the difference in driving behavior caused by different travel purposes significantly affected the fundamental diagrams, such as morning and evening peak hours, weekdays and holidays. The traffic capacity of the same road link in the morning peak hours is 3.47% higher than that in the evening peak hours. The traffic capacity of the same road link on holidays is 4.73% higher than that on weekdays. MFD models considering the road type, morning and evening peak hours, weekdays and holidays improve the flow measurement accuracy by 6.51% compared with that considering only road type.

(4) The case study shows that based on the traffic flow pattern clustering and the recognition method proposed in this paper, the traffic flow pattern is identified quickly, and the whole sample traffic flow on the urban road network is calculated quickly by speed data. The dynamic emissions on the road network are further acquired.

## Figures and Tables

**Figure 1 ijerph-19-16524-f001:**
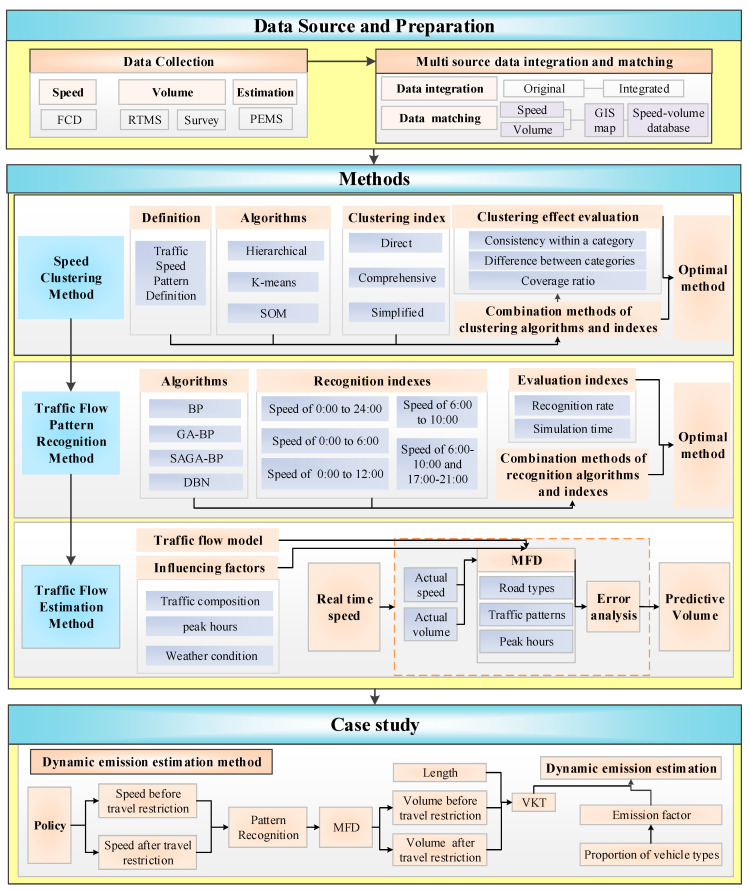
Methodology flow chart.

**Figure 2 ijerph-19-16524-f002:**
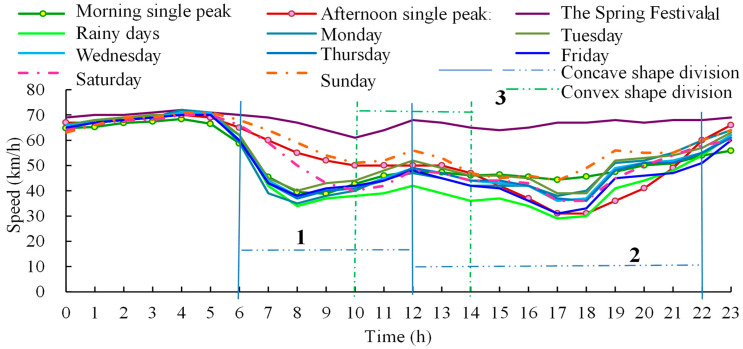
Geometric characteristics of traffic flow speed curves.

**Figure 3 ijerph-19-16524-f003:**
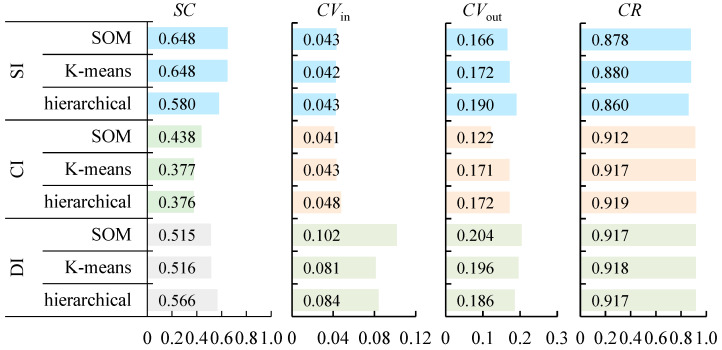
Evaluation of clustering methods under different indexes with different colors.

**Figure 4 ijerph-19-16524-f004:**
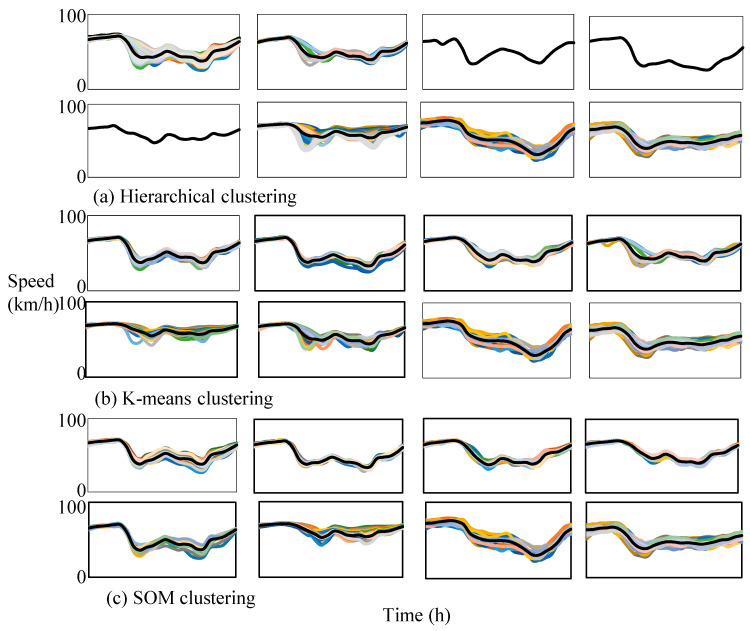
The categories of *DI* based on different algorithms (curves with different colors mean different days).

**Figure 5 ijerph-19-16524-f005:**
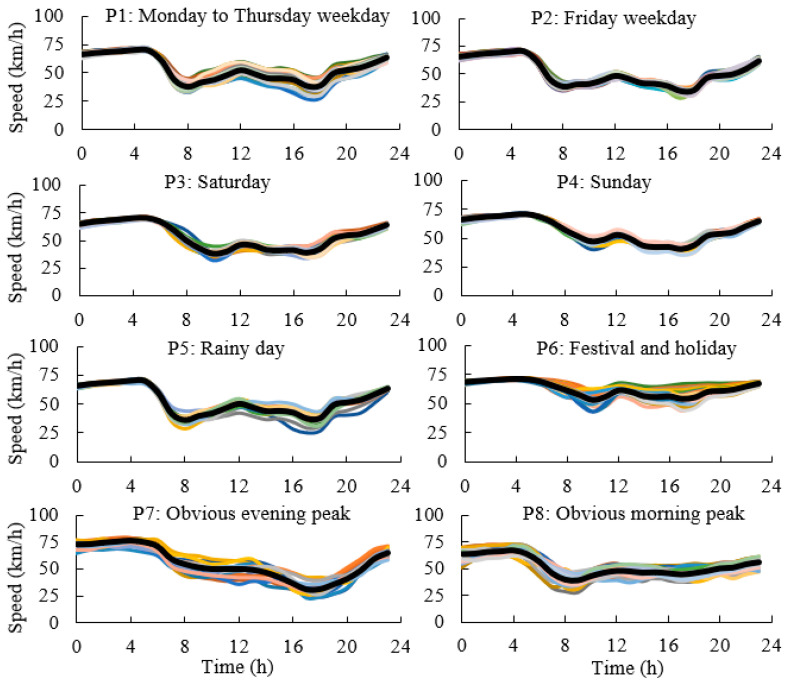
Clustering of expressway speed (curves with different colors mean different days).

**Figure 6 ijerph-19-16524-f006:**
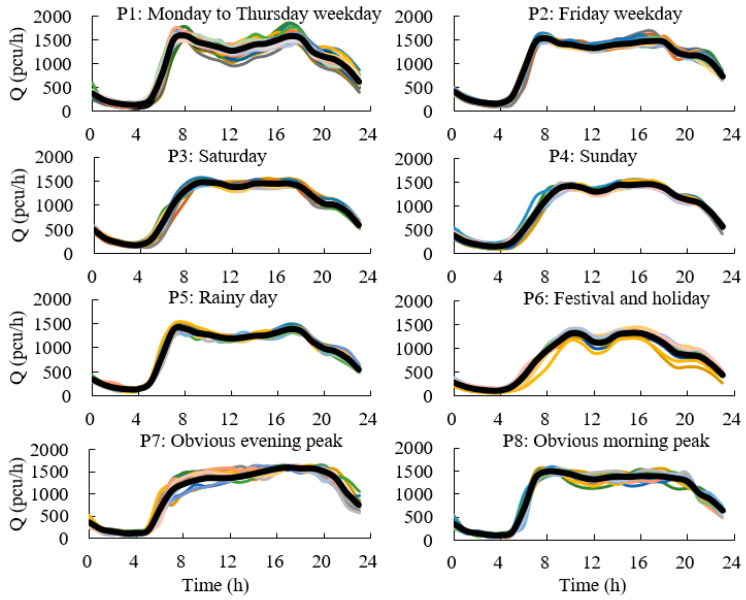
Clustering of expressway volume (curves with different colors mean different days).

**Figure 7 ijerph-19-16524-f007:**
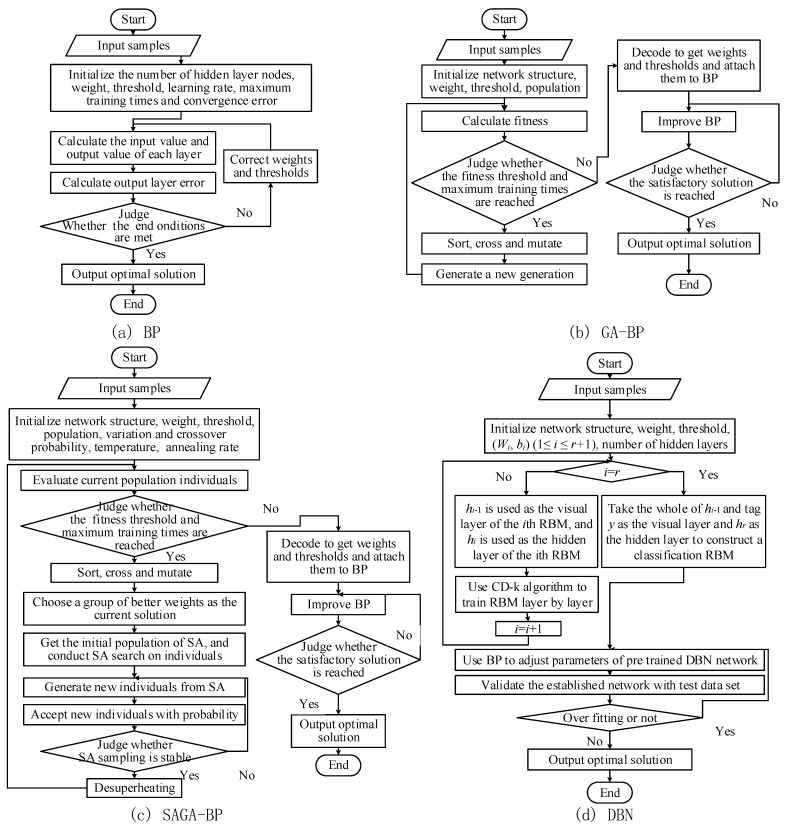
The flow chart of each algorithm.

**Figure 8 ijerph-19-16524-f008:**
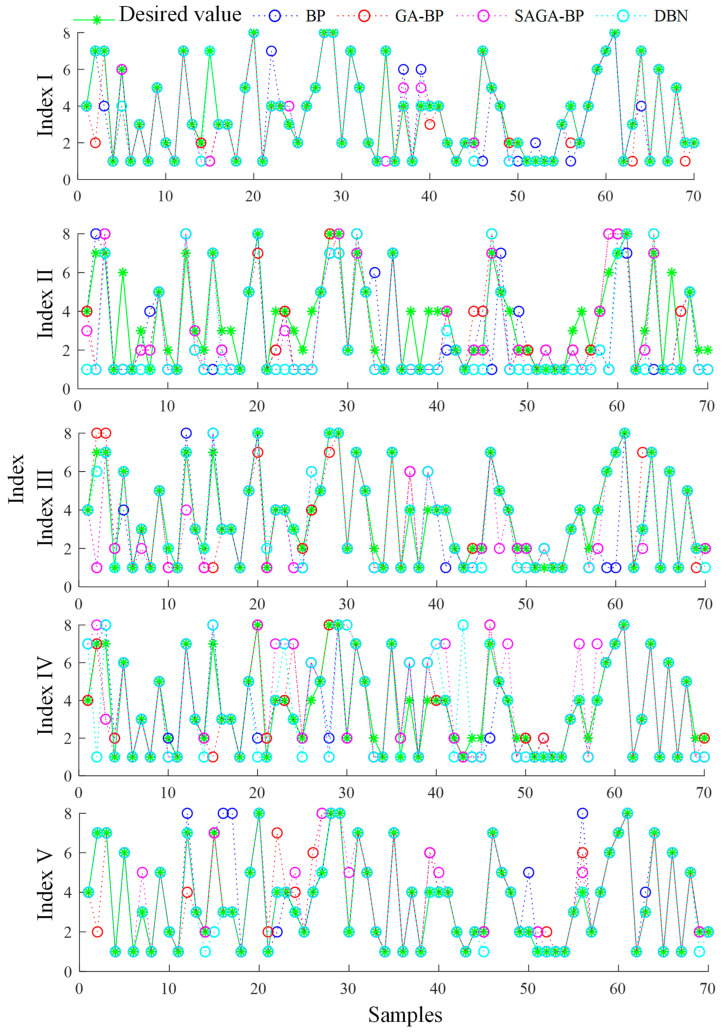
Pattern recognition results of different algorithms based on each index on an expressway.

**Figure 9 ijerph-19-16524-f009:**
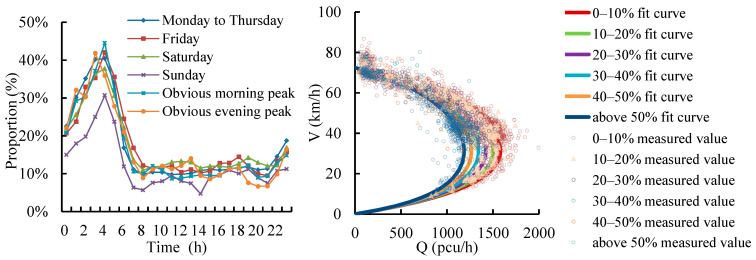
Average speed-flow model under different large-vehicle proportions.

**Figure 10 ijerph-19-16524-f010:**
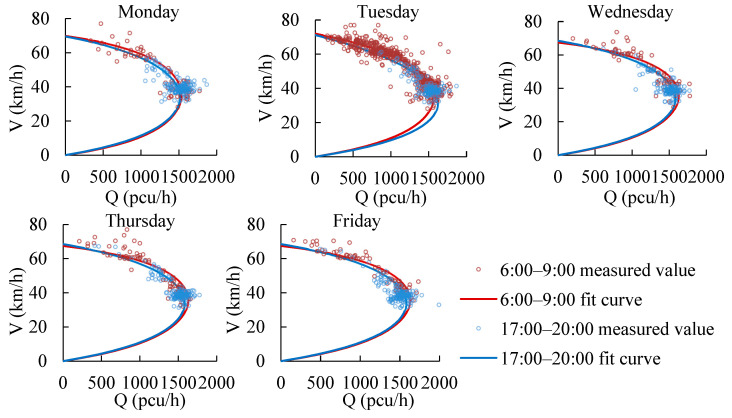
Average speed-flow models of peak hours on different weekdays on expressways.

**Figure 11 ijerph-19-16524-f011:**
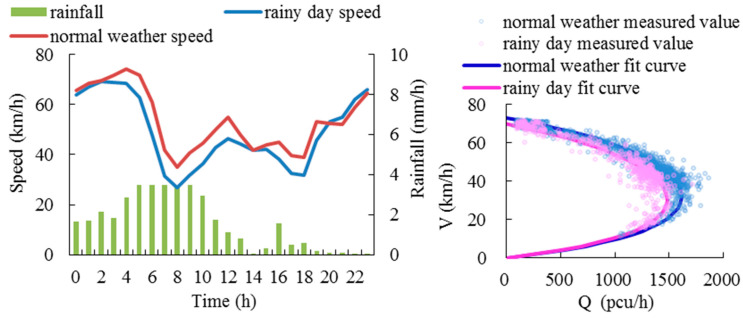
Fundamental diagrams of rainy days and normal days.

**Figure 12 ijerph-19-16524-f012:**
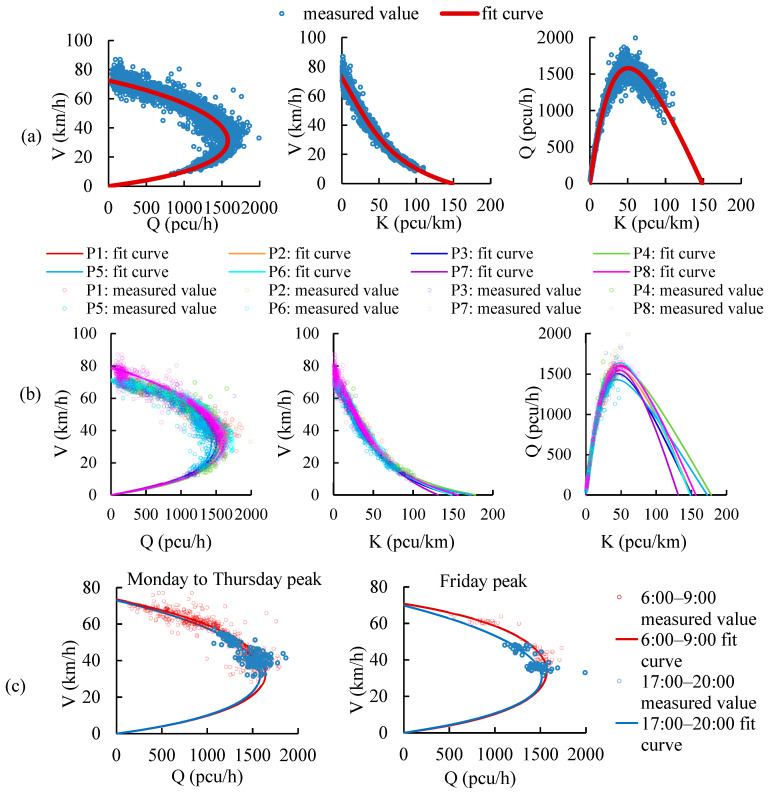
Fundamental diagram models of different proposals on expressways: (**a**) Proposal I, (**b**) Proposal II, (**c**) Proposal III.

**Figure 13 ijerph-19-16524-f013:**
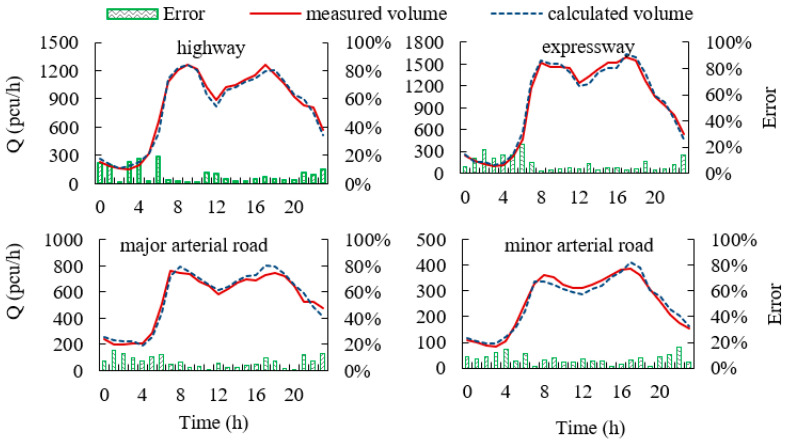
Measured volume and calculated volume curves on different road types.

**Figure 14 ijerph-19-16524-f014:**
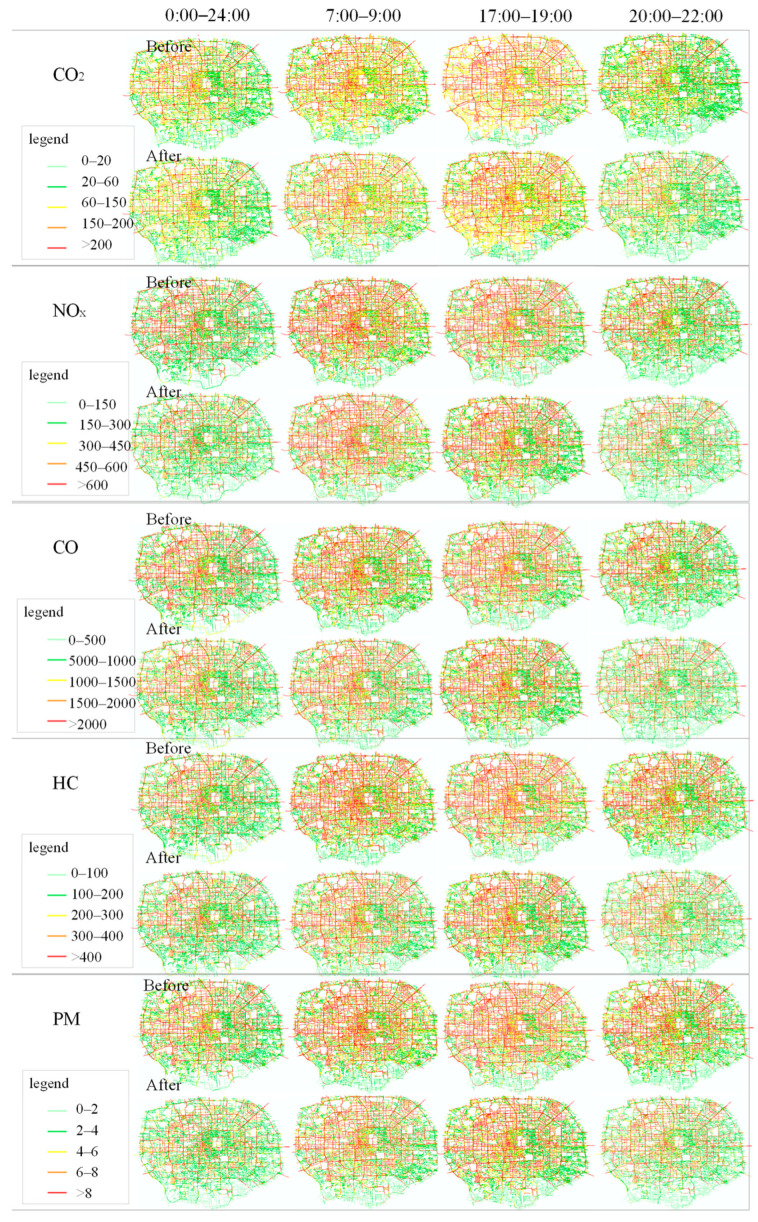
Emission intensity of the road network in different periods.

**Table 1 ijerph-19-16524-t001:** Meaning of *CI*.

Index	*CI*	Index	*CI*
X1	maximum speed	X11	average speed from 0:00–24:00
X2	minimum speed	X12	speed variance from 0:00–24:00
X3	ratio of the maximum and the minimum speed	X13	speed standard deviation from 0:00–24:00
X4	average speed from 0:00–6:00	X14	kurtosis from 6:00–12:00
X5	average speed from 20:00–23:00	X15	skewness from 6:00–12:00
X6	average speed from 10:00–12:00	X16	kurtosis from 12:00–22:00
X7	average speed from 2:00–4:00	X17	skewness from 12:00–22:00
X8	average speed of morning peak	X18	kurtosis from 10:00–14:00
X9	average speed of evening peak	X19	skewness from 10:00–14:00
X10	ratio of the speed of morning peak and evening peak		

**Table 2 ijerph-19-16524-t002:** Meaning of *SI*.

Composition	First	Second	Third	Fourth	Fifth
Feature	Average Feature	High Speed Feature	Concave Feature	Convex Feature	Traffic Flow and Geometric Feature
Index	X11	X10	X16	X18	X1
	X9	X4	X14	X10	X4
	X7	X1	X10	X8	X18
	X2	X15	X6	X19	X3
	X6	X14	X13	X15	X12

**Table 3 ijerph-19-16524-t003:** Parameters of algorithms.

Algorithm	Parameter Design				
Parameter of BP	Name	Hidden layer node	Maximum epochs	Learning rate	Accuracy
	Value	$m=\sqrt{n+l}+\alpha$	100	0.11	0.001
Parameter of GA	Name	Population size	Mutation probability	Crossover probability	Evolutionary algebra
	Value	50	0.05	0.4	200
Parameter of SA	Name	Initial temperature	Annealing speed	Iterations	
	Value	1000	0.85	100	

**Table 4 ijerph-19-16524-t004:** Recognition rates and simulation times of different algorithms based on each index.

Road Type	Algorithm	Recognition Rate of Indexes (%)					Simulation Time (s) of Indexes				
		I	II	III	IV	V	I	II	III	IV	V
Highway	BP	78.79	62.12	66.67	81.82	77.27	5.92	3.56	5.00	3.95	3.80
	GA-BP	89.39	50.00	66.67	89.39	89.39	0.44	1.11	1.34	1.27	1.16
	SAGA-BP	83.33	62.12	72.73	87.88	89.39	0.39	1.09	0.77	1.22	1.06
	DBN	93.94	66.67	77.27	65.15	80.30	0.07	0.07	0.07	0.08	0.07
Expressway	BP	82.86	41.43	72.86	72.86	88.57	2.98	4.59	4.34	4.14	4.75
	GA-BP	90.00	47.14	75.71	72.86	84.29	0.41	1.03	1.27	1.08	1.23
	SAGA-BP	90.00	52.86	72.86	61.43	85.71	0.34	1.31	1.31	1.19	0.38
	DBN	94.29	44.29	80.00	64.29	94.29	0.07	0.07	0.07	0.07	0.10
Major Arterial Road	BP	87.62	49.52	82.86	78.10	92.38	3.27	3.86	3.58	3.58	2.91
	GA-BP	89.52	50.48	82.86	83.81	88.57	0.44	1.03	0.31	0.31	0.22
	SAGA-BP	90.48	56.19	88.57	82.86	92.38	0.42	1.06	0.33	0.33	0.27
	DBN	96.19	55.24	88.57	77.14	94.29	0.07	0.11	0.10	0.10	0.09
Minor Arterial Road	BP	77.78	61.73	74.07	71.61	88.89	5.97	4.39	4.14	4.27	2.94
	GA-BP	87.65	54.32	77.78	74.07	92.59	0.42	1.19	1.53	1.25	1.27
	SAGA-BP	95.06	59.26	66.67	72.84	82.72	0.42	1.05	1.64	1.17	0.52
	DBN	87.65	59.26	83.95	77.78	95.06	0.08	0.08	0.08	0.08	0.07
Optimal Algorithm Recognition Rate		94.87	59.36	82.45	80.96	93.26	0.08	0.08	0.08	0.08	0.08

**Table 5 ijerph-19-16524-t005:** Traffic flow characteristic parameters of different proposals on expressways.

Parameter		Capacity (pcu/h)	Free Flow Speed (km/h)	Critical Speed (km/h)	Blocking Density (pcu/km)
Parameters of Proposal I					
Value		1580	72.5	31.2	148.5
Parameters of Proposal II					
Pattern 1		1535	71.2	32.6	149.4
Pattern 2		1525	71.1	31.8	150.6
Pattern 3		1575	71.4	33.2	170.2
Pattern 4		1590	71.3	32.8	178.2
Pattern 5		1430	70.2	32.1	148
Pattern 6		1615	71.8	33.8	179
Pattern 7		1550	79.8	33.2	148
Pattern 8		1600	78.5	33.2	156.5
Parameters of Proposal III					
Mon. to Thur. peak	AM	1640	73.5	32.9	150.2
	PM	1580	72.9	31.9	148.6
Fri. peak	AM	1570	70.8	33.6	152.1
	PM	1515	69.8	30.2	136.2

**Table 6 ijerph-19-16524-t006:** Error between the measured flow and calculated flow of the three proposals.

Road Type	Proposal I	Proposal II	Proposal III	Proposal III Lower than Proposal I	Proposal II Lower than Proposal I
Highway	15.76%	11.82%	9.70%	6.06%	3.94%
Expressway	12.65%	8.88%	7.64%	5.01%	3.77%
Major arterial road	14.45%	11.09%	8.58%	5.87%	3.36%
Minor arterial road	14.35%	8.74%	7.81%	6.54%	5.61%

**Table 7 ijerph-19-16524-t007:** The emissions of each pollutant before and after the implementation of the traffic restriction policy in different periods.

Pollutant (ton)	Time	0:00–24:00	6:00–23:00	2:00–4:00	7:00–9:00	10:00–12:00	17:00–19:00	20:00–22:00
CO_2_	Before	38,780.98	36,817.12	527.75	4726.34	4061.09	4340.77	2683.81
	After	35,734.09	34,144.84	428.19	4309.21	3774.88	4030.48	2315.86
	Change	−7.86%	−7.26%	−18.87%	−8.83%	−7.05%	−7.15%	−13.71%
CO	Before	546.71	522.25	6.61	67.31	56.63	61.27	35.06
	After	517.55	463.65	5.97	62.77	54.01	58.11	32.29
	Change	−5.33%	−5.11%	−9.77%	−6.74%	−4.63%	−5.15%	−7.91%
NO_X_	Before	144.50	135.96	2.94	17.64	15.31	16.07	8.97
	After	122.22	118.05	1.76	15.24	13.35	13.96	5.28
	Change	−15.42%	−13.18%	−40.17%	−13.64%	−12.79%	−13.12%	−41.15%
HC	Before	100.97	96.56	1.17	12.37	10.47	11.51	6.49
	After	95.50	91.56	1.05	11.52	9.98	10.93	5.97
	Change	−5.42%	−5.18%	−10.34%	−6.92%	−4.64%	−5.10%	−7.97%
PM	Before	2.09	1.95	0.05	0.25	0.22	0.24	0.14
	After	1.61	1.57	0.02	0.20	0.18	0.19	0.06
	Change	−22.84%	−19.78%	−53.44%	−19.80%	−19.71%	−18.97%	−56.13%

**Table 8 ijerph-19-16524-t008:** The emissions intensity before and after the implementation of the traffic restriction policy in different periods.

Pollutant (kg/km/h)	Time	0:00–24:00	6:00–23:00	2:00–4:00	7:00–9:00	10:00–12:00	17:00–19:00	20:00–22:00
CO_2_	Before	163,130.88	134,208.37	14,588.51	172,638.51	165,367.55	198,057.24	145,382.73
	After	154,186.954	126,115.58	10,516.21	167,256.72	156,215.96	185,152.65	136,193.86
	Change	−5.50%	−6.03%	−27.91%	−3.12%	−5.53%	−6.52%	−6.32%
CO	Before	624.292	518.83	82.20	665.68	647.79	743.56	545.60
	After	553.772	450.94	33.27	613.47	578.75	659.00	466.70
	Change	−11.48%	−13.08%	−59.52%	−7.84%	−10.66%	−11.37%	−14.46%
NO_X_	Before	2378.936	1945.46	169.02	2528.96	2396.91	2927.59	2095.76
	After	2300.642	1877.75	145.31	2498.63	2318.82	2793.70	2014.31
	Change	−3.28%	−3.48%	−14.03%	−1.20%	−3.26%	−4.57%	−3.89%
HC	Before	434.278	355.01	29.92	462.06	436.31	528.26	389.75
	After	419.802	342.40	25.23	455.43	421.85	504.88	374.45
	Change	−3.33%	−3.55%	−15.68%	−1.43%	−3.31%	−4.43%	−3.93%
PM	Before	9.24	7.75	1.52	9.65	9.63	10.75	8.42
	After	7.5	6.09	0.40	8.21	7.88	8.79	6.53
	Change	−19.05%	−21.44%	−73.76%	−14.89%	−18.16%	−18.27%	−22.50%

## Data Availability

Some or all data and models that support the findings of this study are available from the corresponding author upon reasonable request.

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
