# Peer review of "A Novel Environment Estimation Method of Whole Sample Traffic Flows and Emissions Based on Multifactor MFD"

_ijerph, 2022, doi:10.3390/ijerph192416524_

Round 1

Reviewer 1 Report

A dynamic estimation of vehicle emissions covering all links of the road network is proposed. Clustering algorithms are used for recognition of traffic flow patterns. Multifactor Microscopic Fundamental Diagram models are developed for estimation traffic flows and emissions, considering different road types, traffic flow patterns and weekday peak hours. And a case study is provided to show change in emissions before and after applying traffic restriction policy of national emission standard.

Several methods are used and compared for clustering and estimation. Quantity and variety of experiments are fine.

Descriptions of methodologies are rather weak.  Algorithms and methods are not clearly explained. It is hard to follow.

In general presentation of the paper and ideas need to be improved. Therefore I would suggest a major revision.

Abstract and introduction could be improved to provide a better summary and explanation of main ideas.

In some parts there is also the need to improve grammar.

In addition to general requirements, some of the minor comments I provide are as follows: 

  • Line 41. There is sth wrong with the sentence “difficult to consistency”
  • Line 37. What it is meant by dynamic estimation is not clear

  • Line 43.  "the lack of dynamic flow covering the whole road network", it s not clear what it is meant by “ lack of dynamic flow”
  • Lines 46-47. use of however is incorrect
  • Line 52. "has developed mature" does not sound good
  • Descriptions in lines 47 to 57 are not well explained. Speed measurement technology is referred to as methodology.
  • Lines 63-64. grammatically incorrect
  • In introduction references should be added for the claims related to existing studies.
  • Line 163. Date types or data types?
  • Line 200.  xi* is used twice
  • Section 4.2, maybe first give pseudo codes of the algorithms and briefly explain them here before talking about clustering index in the next section.
  • Section 4.3, explain what is a clustering index before explaining index design.
  • Details of k-means algorithm such as approximation behaviours, stopping conditions or other parameters are not explicitly described. Maybe those can be explained in section 4.2
  • SOM neural network clustering method is also not explained clearly.
  • Line 315. Does this sentence mean SOM is preferred over K-means for clustering? If so reasons should be explained.
  •  

Reviewer 2 Report

A Novel Environment Estimation Method of Whole Sample 2 Traffic Flows and Emissions based on Multifactor MFD

Vehicle emissions seriously affect air environment and public health. The dynamic estimation method of vehicle emissions has always been the bottleneck of air quality simulation, for there is a lack of an effective method to estimate whole sample dynamic traffic volumes covering all links of the road network. A novel estimation method of whole sample traffic flows and emissions based on multifactor MFD (Macroscopic Fundamental Diagram) is proposed in this paper. The clustering and recognition methods of traffic flow patterns are constructed based on neural network and deep-learning algorithms. Multifactor MFD models are developed for the estimation of whole sample traffic flows and emissions, considering different road types, traffic flow patterns and weekday peak hours. The results show that, traffic flow patterns are clustered efficiently by the Self Organizing Maps (SOM) algorithm combining with the direct time-varying speed index, which describe 91.7% traffic flow states of urban roads. The pattern recognition rate of the deep belief network (DBN) algorithm is 92.1% based on the speeds of the whole day and peak hours. Multifactor MFD models estimate the whole sample traffic flows with high accuracy, the relative error of which 8.43% compared with the measured data. The case study shows that the vehicle emissions are evaluated dynamically based on the novel estimation method proposed in this paper, which is conducive to the coordinated treatment of air pollution.

I clearly read whole manuscript and understand the issues covered in this research is highly valuable for Environment Estimation Method of Whole Sample Traffic Flows and Emissions based on Multifactor MFD.

1. The literature should be updated with research outcomes on par with present scenario.

2. Authors only covered emissions of CO2, it is important that authors should study emissions of SOX, NOX, and COx data along with CO2 emissions.

3. I found authors studies weekdays and other days, but lot of emissions will be accumulated in the air during nights, but there is no collection reports on after 9pm to before 6 am. If the date included it will be meaningful.

Overall, the whole manuscript is strongly narrated including English language. I recommend for the publication.

Result: Minor Revision.

Best

Ravi Kumar Chidrala,

Changwon National University

S. KoreA Novel Environment Estimation Method of Whole Sample 2 Traffic Flows and Emissions based on Multifactor MFD

Vehicle emissions seriously affect air environment and public health. The dynamic estimation method of vehicle emissions has always been the bottleneck of air quality simulation, for there is a lack of an effective method to estimate whole sample dynamic traffic volumes covering all links of the road network. A novel estimation method of whole sample traffic flows and emissions based on multifactor MFD (Macroscopic Fundamental Diagram) is proposed in this paper. The clustering and recognition methods of traffic flow patterns are constructed based on neural network and deep-learning algorithms. Multifactor MFD models are developed for the estimation of whole sample traffic flows and emissions, considering different road types, traffic flow patterns and weekday peak hours. The results show that, traffic flow patterns are clustered efficiently by the Self Organizing Maps (SOM) algorithm combining with the direct time-varying speed index, which describe 91.7% traffic flow states of urban roads. The pattern recognition rate of the deep belief network (DBN) algorithm is 92.1% based on the speeds of the whole day and peak hours. Multifactor MFD models estimate the whole sample traffic flows with high accuracy, the relative error of which 8.43% compared with the measured data. The case study shows that the vehicle emissions are evaluated dynamically based on the novel estimation method proposed in this paper, which is conducive to the coordinated treatment of air pollution.

I clearly read whole manuscript and understand the issues covered in this research is highly valuable for Environment Estimation Method of Whole Sample Traffic Flows and Emissions based on Multifactor MFD.

1. The literature should be updated with research outcomes on par with present scenario.

2. Authors only covered emissions of CO2, it is important that authors should study emissions of SOX, NOX, and COx data along with CO2 emissions.

3. I found authors studies weekdays and other days, but lot of emissions will be accumulated in the air during nights, but there is no collection reports on after 9pm to before 6 am. If the date included it will be meaningful.

Overall, the whole manuscript is strongly narrated including English language. I recommend for the publication.

Result: Minor Revision.

Best

Ravi Kumar Chidrala,

Changwon National University

S. Kore
